# The Role of Gender and Familiarity in a Modified Version of the Almeria Boxes Room Spatial Task

**DOI:** 10.3390/brainsci11060681

**Published:** 2021-05-22

**Authors:** Alessia Bocchi, Massimiliano Palmiero, Jose Manuel Cimadevilla Redondo, Laura Tascón, Raffaella Nori, Laura Piccardi

**Affiliations:** 1Department of Psychology, Sapienza University of Rome, 00185 Rome, Italy; laura.piccardi@uniroma1.it; 2Biotechnological and Applied Clinical Sciences, University of L’Aquila, 67100 L’Aquila, Italy; massimiliano.palmiero@univaq.it; 3Health Research Center and Department of Psychology, University of Almería, 04120 Almería, Spain; jcimadev@ual.es; 4Department of Psychology, University of Cordoba, 14014 Cordoba, Spain; ltascon@uco.es; 5Department of Psychology, University of Bologna, 40126 Bologna, Italy; raffaella.nori@unibo.it; 6Cognitive and Motor Rehabilitation and Neuroimaging Unit, IRCCS Fondazione Santa Lucia, 00179 Rome, Italy

**Keywords:** spatial navigation, sex differences, environment familiarity, spatial learning, spatial knowledge, route, survey, virtual environments

## Abstract

Individual factors like gender and familiarity can affect the kind of environmental representation that a person acquires during spatial navigation. Men seem to prefer relying on map-like survey representations, while women prefer using sequential route representations. Moreover, a good familiarity with the environment allows more complete environmental representations. This study was aimed at investigating gender differences in two different object-position learning tasks (i.e., Almeria Boxes Tasks) assuming a route or a survey perspective also considering the role of environmental familiarity. Two groups of participants had to learn the position of boxes placed in a virtual room. Participants had several trials, so that familiarity with the environment could increase. In both tasks, the effects of gender and familiarity were found, and only in the route perspective did an interaction effect emerge. This suggests that gender differences can be found regardless of the perspective taken, with men outperforming women in navigational tasks. However, in the route task, gender differences appeared only at the initial phase of learning, when the environment was unexplored, and disappeared when familiarity with the environment increased. This is consistent with studies showing that familiarity can mitigate gender differences in spatial tasks, especially in more complex ones.

## 1. Introduction

When it is necessary to remember an object’s location in the environment, humans can refer to their own position using an “egocentric frame of reference” (e.g., ‘the fountain is at my left’), or refer to the spatial and configurational properties of such objects, using an “allocentric frame of reference” (e.g., ‘the fountain is at the left of the shop’) [1,2]. These two frames of reference lead to acquisition of two different types of spatial representations: one named *route*, and the other named *survey*, respectively [3,4,5,6,7,8]. The *route* representation is characterized by an egocentric perspective, referring to the sequential aspects of navigation, such as the connection between different landmarks and paths. The *survey* representation is organized as a map and is characterized by an external perspective in spite of the individual’s position in the environment [5].

Actually, the kind of environmental representation that a person may acquire can be affected by several factors, external and internal. Examples of external environmental factors are the presence of landmarks [9,10,11], the environmental complexity [12] or high-impact environmental changes [13]. Examples of internal factors, i.e., those that belong to the traveler [12,14], are environmental familiarity (the degree of environmental knowledge [15,16,17], expertise [18,19], cognitive style [20,21,22,23,24], gender [25,26,27,28,29], and emotional states [30], among others. Among internal factors, gender and familiarity, which the present paper is focused on, play a crucial role.

Concerning gender, men and women pay attention to different environmental features and cues while navigating, eliciting different strategies to orient themselves through the world.

Men prefer using ‘‘survey strategies’’ which rely on global reference points and seem to acquire high-order spatial knowledge more quickly, which is crucial to choose the best strategy to reach a goal [20,31]. Women, instead, rely on landmarks (environmental objects that people use as reference points, [11]) and procedural ‘‘route strategies’’, that rely on how to get from a place (or a landmark) to the other [32,33,34,35,36,37].

This preference in strategy use and environment exploration can account for men’s advantage in navigation tasks, as suggested by the studies of Munion et al. [38] and Boone et al. [39].

In the study of Boone et al. [39], where men and women had to reach a goal with and without distal landmarks in a virtual environment, sex differences were found in measures of both route selection and navigation efficiency. Males were more likely to take shortcuts, and reached their goal location faster than females, while females were more likely to follow learned routes and wander, sometimes finding the goal as a result of wandering. Munion et al. [38] showed that men and women produced different wayfinding behaviors which, in turn, predicted differences in navigational success even when they belonged to a population with high navigation competence (West Point cadets).

Such a difference in the choice of spatial strategy is also evident at a neural level: Grön et al. [40] found that during the same spatial learning task in a virtual environment, different brain areas were active in men and women.

According to Siegel and White’s model [3], familiarity with the navigational environment allows switching from the lower (landmark or route) to the higher (survey) level of spatial knowledge. In other words, the higher the familiarity, the more complete the environmental mental representation [41,42]. For this reason, a good familiarity with the environment allows performance of complex navigational tasks even in individuals with poor navigation abilities [15]. Indeed, familiarity increases the ability to remember the presence of points of reference and their positions in the environment (e.g., [43]), but can also influence an individual to choose a specific frame of reference to represent a real-world environment [44,45]. When the environment is familiar, people are more prone to use an allocentric frame of reference.

The effect of familiarity is so strong that it can affect other individual factors, such as gender [7,15,16,46,47,48]. Once a good familiarity with the environment has been acquired, gender differences in spatial abilities may be minimal (e.g., [49]) or can even disappear (e.g., [15,16,44]).

Despite the wide number of studies assessing the role of familiarity and gender differences in spatial navigation, the interaction between these two factors is still debated. In particular, it is still discussed how gender and familiarity with the environment can affect spatial abilities when different modalities of environmental knowledge acquisition are required (i.e., route or survey).

In this study, two independent samples performed the Walking Space Boxes Room Task that relates to the Route task based on the Boxes Room task [50], or the nonwalking space Boxes Room Task that relates to the Survey task, which is a modified version of the Boxes Room task [51].

The Boxes Route task [50] requires learning the position of a series of boxes in a walking virtual environment by moving with a first-person perspective around the room, whereas the Boxes Survey Task [51] requires learning the position of a series of boxes dis-placed in the same virtual room of the Route task but by viewing the room from different perspectives without the possibility to change the point of view or walk through the envi-ronment.

Moreover, while the Boxes Route Task allows a different navigation experience for each participant, the experience in the Boxes Survey Task is the same for each participant. This specificity of the task contributes even more to promoting the presence of individual differences.

One important feature of both tasks is that the virtual room was enriched with environmental cues such as windows and doors. This aspect represents a critical feature of the task and an innovation with respect to other tasks, as it provides the possibility of using multiple topographical cues to facilitate performance, which is closer to the navigational/orientation context of the original hippocampal place studies by O’Keefe and Nadel [52]. These environmental cues were placed to let participants take advantage of them, trial by trial, to improve performance.

In other words, given that the environment was always the same and the landmarks visible, the level of familiarity with environment increased with each trial. As a consequence, the visuo-spatial working memory load decreased [15].

As in [53], the task began with a poor level of familiarity/difficulty and then continued with a middle and a high level of difficulty and familiarity.

Given that gender differences can be found in many spatial tasks (for a review see [54,55], we expected to find gender differences in terms of accuracy in both the Route and Survey tasks. Indeed, men and women pay attention to different environmental features and cues while navigating, and consequently elicit different strategies to spatial orient themselves in the world. For instance, men rely on global reference points and configurational strategies, while females more often use landmarks and procedural strategies when finding how to get from place to place [37,39]. In particular, the study of Boone et al. [39] evidenced that men use shortcuts and take less time to solve a navigation task, and these differences can be explained by the different use of navigational strategies.

In line with other studies [3,7,15,16], it was expected that as familiarity with the environment increased, i.e., as participants took advantage of environmental cues to find the target boxes, gender differences in spatial learning may be reduced in both tasks, regardless of the increasing level of difficulty.

## 2. Materials and Methods

### 2.1. Participants

Two independent samples were enrolled. The first sample underwent only the Route task, and the second sample underwent only the Survey task. The first sample, which underwent the Route task, encompassed 104 participants, while the second sample, which underwent the Survey task, encompassed 109 individuals. Participants were enrolled from the University of L’Aquila (L’Aquila, Italy) and from University of Almeria (Almeria, Spain) (for details see Table 1). All participants were college students from different fields of study. The inclusion criterion (the same for Italy and Spain) was no history of neurological/psychiatric diseases (including substance abuse or dependence). Before taking part in the study, participants of both samples filled in a questionnaire in which they self-reported any previous/current neurological or psychiatric disorder. The questionnaire included a specific question on spatial orientation disorders to make sure that none of the participants suffered from Developmental Topographical Disorientation (a neurodevelopmental disorder that may affect healthy individuals impairing their ability to learn new environments, to retrieve environmental information, as well as to recognize landmarks [17,45,56]). According to the Declaration of Helsinki, before the testing phase and after a full explanation of the protocol and of the noninvasiveness of the study, a written informed consent was obtained from all participants included in the study. The study was approved by the Ethics Committee of the University of Almeria (Ethical approval number: UALBIO2015/012).

The two samples were composed as follows (see also Table 1).

Route task: one hundred and four college students (mean age 23.5 ± 16.02; 55 males and 49 females). Age was comparable between groups (t(102) = −0.391; *p* = 0.146).Survey task: one hundred and nine college students (mean age 21.85 ± 2.60; 52 males and 57 females). Age was comparable between groups (t(107) = 1.787; *p* = 0.077).

### 2.2. Procedure

For both samples, participants were taken to a quiet room where they were assessed individually. They were asked to sit on a comfortable chair in front of the experimenter. At the beginning of each experimental session, the experimenter gave a leaflet and explained to the participants all the information regarding the aim, procedure, risks and advantages of the study. Participants were also informed about their rights and about the possibility to leave the study at any time they wished. Furthermore, the experimenter invited the participants to express any doubts or questions they might have on the study. After explaining how their personal data would be processed, they signed the informed consent, gave the authorization to use their personal data and provided a brief medical history. Then, participants performed the virtual navigational task. Given that the experimental tasks, in their original versions (see [50] for the Route task and [51] for the Survey task) involved a virtual environment and specific tools (a joystick for the Route task and a mouse for the Survey task), before performing the experimental tasks participants underwent a preliminary session to ensure they were sufficiently confident with the tools and the virtual environment. Participants were introduced to a simplified version of the virtual environment (see below for details), with only two boxes. For the Route task, participants had the opportunity to try the joystick and to move around the room. For the Survey task, participants tried to open the boxes by using the mouse. When participants declared confidence with the procedure, the experimental task was presented.

As in [53], for both tasks, three levels of familiarity were used, each consisting of 10 trials: low, medium and high. The inter-trial interval was 5 s. In order to avoid a plateau effect across the trials, the number of reward boxes increased from the first to the third level of familiarity. Specifically, the number of reward boxes to find were three in the first 10 trials; five in the second 10 trials, seven in the last 10 trials (see also [53]).

The two tasks are described in detail below.

#### 2.2.1. The Route Task

Participants were required to use the joystick to move around a three-dimensional room with 16 brown boxes symmetrically distributed on the floor (see Figure 1). The aim of the task was to find the position of several reward boxes. For each trial, the box turned blue when it was reached. Then, by pressing a button, it was possible to open the box. If a reward box was opened, it turned green, and a pleasant melody sounded. If a wrong box was opened, it turned red, and an unpleasant tone sounded. During the same trial, the opened boxes remained green or red until the participant found all the reward boxes or until the maximum trial duration (150 s) was reached. The reward boxes remained in the same locations during the experiment; accordingly, participants could improve performance from trial to trial. There were several stimuli in the room that disambiguated spatial locations, including several pictures, a window, and a door. Participants were asked to find the reward boxes as quickly as possible and avoid opening the wrong boxes. There were four different starting positions, which changed randomly between trials. Subjects were not informed about the possible spatial strategies or the position of the reward boxes.

This task was characterized by an egocentric frame of reference, that is, it required using the position of the moving self to locate objects in the environment. By allowing movement in the environment through the joystick, such a task permitted individuals to detect the reward boxes by the relative position of the self, leading to a route knowledge.

The route task was implemented in MATLAB using Cogent 2000 (Well- come Laboratory of Neurobiology, UCL, London, www.vislab.ucl.ac.uk/cogent.php; accessed on 10 September 2016).

#### 2.2.2. The Survey Task

In the Survey task, the same virtual room with 16 boxes symmetrically distributed on the floor was showed from the above (as in a bird’s eye view, see Figure 2). Participants were requested to use the mouse to click on the boxes and find the position of several reward boxes.

For each trial, the box turned blue when the mouse pointer was on it; then, by pressing the mouse, it was possible to open the box. If a reward box was opened it turned green and a pleasant melody sounded. If a wrong box was opened it turned red and an unpleasant tone sounded. During the same trial, the opened boxes remained green or red until the participant found all the reward boxes or until the maximum trial duration was reached (150 s). The reward boxes remained in the same locations during the experiment; accordingly, participants could improve performance from trial to trial. There were several stimuli in the room that disambiguated spatial locations, including several pictures, a window and a door.

Participants were asked to find the reward boxes as quickly as possible and avoid opening the wrong boxes. The room was shown changing the point of view (the four sides of the room: north, south, east, and west), which changed randomly between trials.

In other words, participants could only see the virtual room from one of the four walls. From that point of view, it was possible for them to see the other three walls with the same landmarks as before and a total of sixteen brown boxes ordered by rows of four. Subjects were not informed about spatial strategies or the position of the reward boxes.

This task was characterized by an allocentric frame of reference, that is, it required use of the position of the objects in the environment to locate other objects. To locate the target boxes, it was necessary to detect their position with respect to the other boxes or use the environmental cues (the door, the paintings on the wall, etc.). In other words, by forbidding rotation of the perspective of the room the task elicited the use of an allocentric frame of reference and led to acquisition of survey knowledge.

The Survey task was implemented in MATLAB using Cogent 2000 (Well- come Laboratory of Neurobiology, UCL, London, www.vislab.ucl.ac.uk/cogent.php; accessed on 10 September 2016).

## 3. Analyses

For both the Route and Survey tasks the number of errors was detected. Note that trial 1 was removed from analyses since performance was at random.

To exclude possible nationality effects, a 2 × 3 mixed ANOVA (nationality × familiarity) was performed, with nationality/group as the between independent variable, familiarity as the within independent variable, and numbers of errors as the dependent variable.

Gender differences in familiarity levels (low, middle and high) were analyzed. A 2 × 3 mixed ANOVA (Gender × Familiarity) was performed for each task with gender as the between independent variable, familiarity level as the within independent variable, and numbers of errors as the dependent variable.

The Bonferroni procedure was used for post hoc comparisons when necessary. A significance level of *p* < 0.05 was used for analysis.

## 4. Results

### 4.1. Effect of Nationality

For the Route task, the ANOVA showed a nonsignificant main effect of ‘nationality’ (F(1, 102) = 2.15, *p* = 0.14, partial eta-squared = 0.02). Spanish and Italians did not differ in the number of errors committed. A significant main effect of ‘familiarity’ (F(2, 204) = 29.43, *p* < 0.001, partial eta-squared = 0.22), with post hoc comparisons (Bonferroni) showed that the condition with low familiarity yielded more errors than both the condition with middle (*p* < 0.001) and high familiarity (*p* < 0.001). No significant difference was found between the condition with middle familiarity and high familiarity (*p* = 0.43). In addition, a nonsignificant effect of interaction ‘nationality × familiarity’ was found (F(2, 204) = 1.01 *p* = 0.36, partial eta-squared = 0.01).

For the Survey task, the ANOVA showed a nonsignificant main effect of ‘nationality’ [F(1, 107) = 0.02, *p* = 0.88, partial eta-squared = 0.000]. Spanish and Italians did not differ in the number of errors committed. A significant main effect of ‘familiarity’ (F(2, 214) = 42.57 *p* < 0.001, partial eta-squared = 0.28), with post hoc comparisons (Bonferroni) showed that the condition with low familiarity yielded more errors than the condition with middle familiarity (*p* < 0.01) and high familiarity (*p* < 0.001). The condition with middle familiarity also yielded more errors than the condition with high familiarity (*p* < 0.001). The interaction ‘nationality × familiarity’ was nonsignificant (F(2, 214) = 0.70; *p* = 0.50; partial eta-squared = 0.006).

### 4.2. Gender Differences: Route Task

The ANOVA showed a significant main effect of gender (F(1, 102) = 11.81, *p* < 0.001, partial eta-squared = 0.10) (See Figure 3). Men committed less errors than women. A significant main effect of ‘familiarity’ (F(2, 204) = 31.39, *p* < 0.001, partial eta-squared = 0.24) with post hoc comparisons (Bonferroni) showed that the condition with low familiarity yielded more errors than both the condition with middle (*p* < 0.001) and high familiarity (*p* < 0.001). No significant difference was found between the condition with middle familiarity and high familiarity (*p* = 0.42). In addition, a significant effect of interaction ‘gender × familiarity’ was found (F(2, 204) = 3.28 *p* < 0.05, partial eta-squared = 0.03). Women made more errors in the condition with low familiarity respect to both the conditions with middle (*p* = 0.001) and high familiarity (*p* = 0.001). No significant differences appeared between the condition with medium and high familiarity (*p* = 1).

In males group participants committed more errors in the condition with low familiarity respect to the condition with middle (*p* < 0.05) and high familiarity (*p* < 0.01). No difference was found between medium and high (*p* = 1).

In addition, women made more errors than men only in the condition with low familiarity (*p* < 0.001).

No difference was found in the condition with middle (*p* = 0.10) and high familiarity (*p* = 0.77) between men and women.

### 4.3. Gender Difference: Survey Task

The ANOVA showed a significant main effect of ‘gender’ (F(1, 107) = 4.32, *p* < 0.05, partial eta-squared = 0.04). Men committed less errors than women. A significant main effect of ‘familiarity’ [F(2, 214) = 42.47, *p* < 0.001, partial eta-squared = 0.28], with post hoc comparisons (Bonferroni) showed that the condition with low familiarity yielded more errors than the condition with middle (*p* < 0.01) and high familiarity (*p* < 0.001). The condition with middle boxes also yielded more errors than the condition with high familiarity (*p* < 0.001). The interaction ‘gender × familiarity’ was nonsignificant (F(2, 214) = 0.05; *p* = 0.96; partial eta-squared: 0.000). (See Figure 4).

There were no significant differences between men and women in all the familiarity levels.

## 5. Discussion

The present paper was aimed at investigating gender differences in two spatial learning tasks, considering two modalities of environmental knowledge acquisition (route and survey) and familiarity with the environment. We found that men outperformed women in both Route and Survey tasks. However, in the Route task, women made more errors only in the first phase of learning (low familiarity condition). Moreover, in the Survey task, an effect was found for familiarity with a decreasing number of errors passing from low to high familiarity.

These results are consistent with the well-established evidence that men outperform women on spatial tasks and navigation performance [54,55]. In particular, the literature suggests that reaching more easily a high-order spatial knowledge allows men to be better in many spatial tasks. Indeed, men are better at learning routes on a map both in the real world and in a virtual environment [57,58], in mentally transforming environmental information and in estimating distances (e.g., [54]). Moreover, with respect to women, they show a higher memory span (i.e., the longest list of items that a person can repeat in correct order immediately after the presentation) [17,25,59,60,61] and better memory of a path from different points of view [62]. Even during navigational tasks in finding a path, men are better and faster, and take shortcuts more easily than women who are more likely to wander or follow an already learned path [39].

However, in the Route task, despite women making more wrong attempts to learn the position of the reward boxes when the environmental familiarity was low (first condition), at the end (high familiarity condition) they were able to reach the same level as the men’s performance. In this vein, Iachini et al. [39] showed that men were faster than women in learning the supra-span sequence. Gender differences, however, disappeared once the sequence had been learned.

Moreover, Piccardi et al. [63] found that during a reorientation task, gender differences emerged only in the learning phase. In addition, they found that gender differences decreased when participants could take their time to repeat the task as many times as needed. Similarly, when men and women were required to learn a path from a map or by observing an experimenter in a real environment, gender differences were not present in the retrieval phase when women were given the necessary time to acquire spatial information [7]. This suggests that gender differences can disappear when spatial learning is well-acquired.

Gender differences were found only in the first condition of the Route task, which was the most challenging condition because the environment was new. Therefore, it may be presumed that participants reached a certain level of familiarity with the environment in the subsequent conditions and performed better, even though the number of boxes increased (five and seven boxes). Indeed, the virtual room was enriched with landmarks, such as a door and paintings on the wall, that participants may have used to quickly find the reward boxes (‘the right boxes were near the window’). When the environment was new, gender difference emerged, but disappeared as familiarity increased. In other words, increasing familiarity with the environment allowed women to narrow the gap with men.

Interestingly, the familiarity effect was not found in the Survey task, where only gender and condition effects emerged. It is possible that the Survey task was too difficult for women, and probably they would need more attempts to reach a performance comparable to men, as in Piccardi et al. [63] in which sex differences disappeared when women had more time to learn. This explanation is in line with data from previous studies that found that subjective learning of environmental material is different between men and women. Generally speaking, women need more time to learn a map and more repetitions to learn a path [63]. Indeed, in Nori et al. [7], when different times and different numbers of repetitions were given to men and women, they did not find any differences between groups despite the difficulty of the navigational task. Furthermore, also in Nori and Piccardi [64], even if women self-assessed as being less able to navigate, when they performed tasks concerning a familiar environment, no differences emerged between groups. It is also possible that women needed a higher level of familiarity than men to reach a comparable performance (that can be translated in more trials) in the Survey task, but not in the Route task. This is probably because in the Route task women could apply a strategy that better fit their ability level, whereas in the Survey task, where a route strategy is less convenient, more repetitions were necessary for women and gender differences may have become more explicit.

Thus, it is possible that the cognitive load of the Survey task played a key role. Indeed, Piccardi et al. [63] found that gender differences emerged only in adverse learning conditions that required strong spatial ability, suggesting that interactions between environmental demands and cognitive processes modulate gender differences. This result was found also by Grön et al. [40], who found a prefrontal cortex activation in women that suggested a working memory load in solving the navigational task, while men used the human navigation network. In this task participants had to recall the environment, the objects contained therein and their relative positions, as well as mentally re-orient a photo based on their memory of the environment (the photo showed in the Survey task had different orientations). It is possible that these mental operations may have increased the cognitive load at a level that nulled the familiarity effect. Instead, unlike the Survey task, in the Route task (where manipulating the representation of the environment could be a good strategy to quickly find the right boxes), if the cognitive requirements were too demanding, one could compensate by moving the joystick around the room to find the right answer. This observation is in line with the Environmental Knowledge Model-EKM, [15], which highlights that the higher the cognitive load, the more important the ability to correctly represent spatial information. Individuals with a more flexible mental representation of the environment perform the navigational tasks better even if they are very difficult. Importantly, this is true regardless of familiarity. Individuals with high navigational abilities are more proficient in solving spatial tasks. Therefore, the EKM model suggests considering not only the mental representation that a person may create, but also the complexity of the spatial task that should be solved, regardless of familiarity.

In the future it will be interesting to conduct more research to clarify all these aspects.

## 6. Conclusions

This study showed that spatial learning tasks in the route and survey modalities of navigation can be affected by gender, with men showing better performance. Importantly, the effect of familiarity modulated gender differences in the route but not in the survey modality. These findings suggest that familiarity with the environment can smooth gender differences when the task is not cognitively demanding, as for the Route task, whereas when the task difficulty increases, as in the Survey task, familiarity is not sufficient to negate gender differences.

In the future it could be interesting to investigate if such differences can be explained considering the strategies used by women and men in the light of previous studies such as [49], which showed that sex difference can be explained in the light of different strategies used by the two sexes [38,39].

## Figures and Tables

**Figure 1 brainsci-11-00681-f001:**
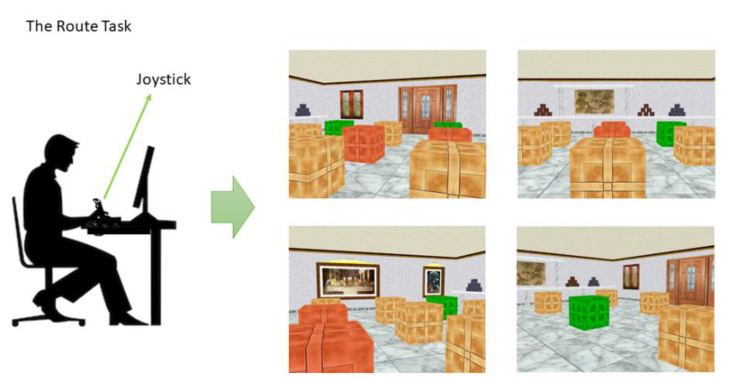
Four examples of the Route Task in the virtual room. Participant can move around the room with a joystick using a first person, egocentric perspective. Note that boxes are brown, and they changed their colour when opened. Green boxes represent right choices whereas red boxes indicate errors.

**Figure 2 brainsci-11-00681-f002:**
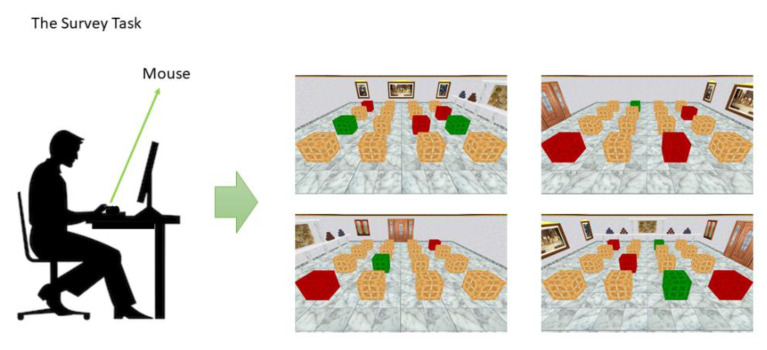
Four examples of the Survey Task in the virtual room. Participant saw the same virtual room as in the Route task, but from different points of view and could not move around. They could use a mouse to select the boxes to open.

**Figure 3 brainsci-11-00681-f003:**
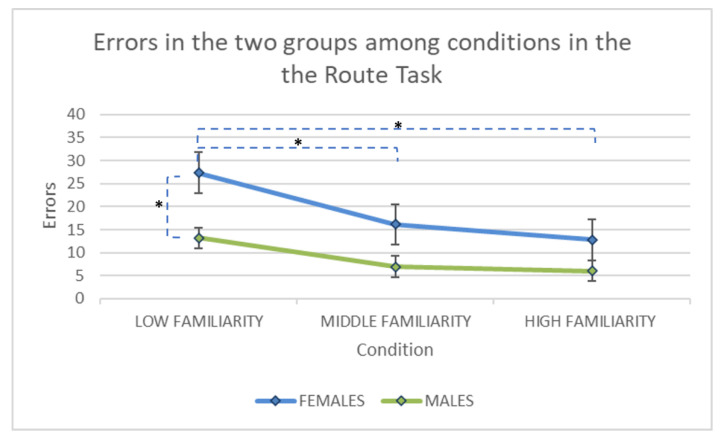
The number of errors made by the two groups (M = male group: blue line; F= female group, green line), among the three conditions (three, five and seven reward boxes) in the Route task. Asterisks indicate the significant differences between groups or conditions within groups.

**Figure 4 brainsci-11-00681-f004:**
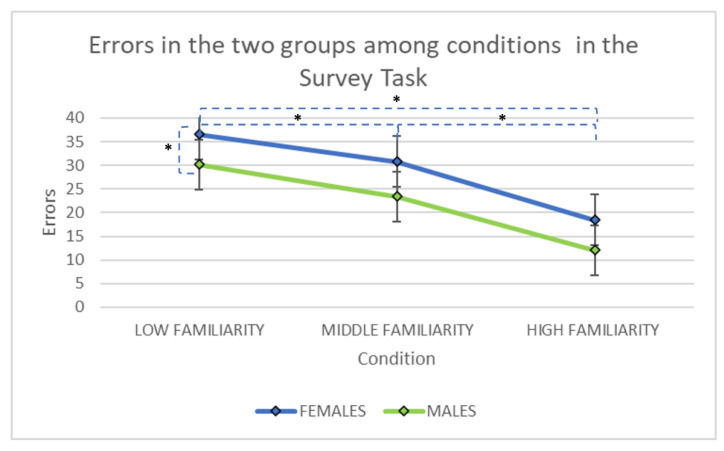
The number of errors made by the two groups (M = male group: blue line; F= female group, green line), among the three conditions (three, five and seven reward boxes) in the Survey task. Asterisks indicate the significant differences between groups or conditions within groups.

**Table 1 brainsci-11-00681-t001:** The demographic features of the samples performing the Route and the Survey task, respectively.

	Males	Females	Italy	Spain
Route task N = 104	55	49	55	49
Survey task N = 109	52	57	63	46

## Data Availability

Not applicable.

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
