# Peer review of "The Role of Gender and Familiarity in a Modified Version of the Almeria Boxes Room Spatial Task"

_brainsci, 2021, doi:10.3390/brainsci11060681_

Round 1

Reviewer 1 Report

MS Review: The role of gender and familiarity in a modified version of the Almeria Boxes Room Spatial Task

General Comments

This was an interesting manuscript which I thoroughly enjoyed reading – thank you.

However, I do feel that there are several issues to address in the manuscript.

  1. I feel that the Introduction could have had a more contemporary research, I have highlighted some such studies, inclusion of this research may have led to a more nuanced consideration of the hypotheses, e.g., gender differences in some conditions, such as early on in the spatial topography learning,
  2. In the Methods, are the original task labels an identification in the earlier use of the methodology (e.g. Tascón et al., 2018) more appropriate? Also, I found the task detail and scoring more informative and clearer to read in the Tascón et al paper,
  3. In the Results section the key variable Gender, is not fully explored, post hoc analyses on gender differences at each level of the two conditions (‘route and ‘survey’) would have been informative and effect sizes with these analyses would have enabled a contrast with the prior literature,
  4. Unless I have completely misunderstood the results I feel that many of the interpretations of the findings in the Discussion are not supported by the results, the authors should consider this.

Specific Comments

L22-23, typo 'difference scan' should be 'differences can',

L32-35, good, very clear description of the concepts,

L56-57, relatively dated references cited here, Boone et al, 2018, Munion et al, 2019 could be usefully considered

L66-68, good use of  Iachini et al 2009 research,  

L72, dated study on gender differences, perhaps more pertinent are the Voyer et al series of meta-analyses, e.g. Voyer et al, 2007 (Gender differences in object location memory) and 2017 (visuospatial working memory, VSWM),

L95-97, thus, as familiarity increases, so does visuospatial working memory load,

L100-101, Lawton chapter is dated and Nazareth et al (2019) meta-analysis may have provided a more nuanced consideration??

L113, 'Route task' in the original Tascon et al study this was labelled as the walking task, 'Survey task' in the original Tascon et al study this was labelled as the non-walking task. I am not entirely convinced labelling the task conditions as ‘route’ and ‘survey’ really captures the essential difference in the tasks as well as the original walking/non-walking labels,

L121-125, a very rigorous exclusion criterion,

L164-167, having 5 and 7 locations to be remembered in phases 2 and 3 places a particularly large demand upon VSWM as the procedure has extensive demands and the basic Corsi task span is 6-6 (e.g Piccardi et al, 2013). This could suggest that VSWM supra-span learning is occurring as the trials and complexity develop and this could be one contribution to individual differences in overall task performance,

179-180, “… There were several stimuli in the room that disambiguated spatial locations, including several pictures, a window and a door.…”  I think that is a critical feature of the task, particularly in the ‘survey’ condition as it takes it away from the CANTAB Spatial Working Memory Procedure, the Corsi, and the Walking Corsi Test (Piccardi) and closer to the navigational/orientation context of the original hippocampal place studies by O'Keefe & Nadel (1979), now participants can use multiple topographical cues to facilitate performance,

L183-184, “Subjects were not informed about the possible spatial strategies or the 183 position of the rewarded boxes.”  This means that first trial score should be discounted in the analysis as it is a predominantly random process at work,

L186-188, “By allowing to move in the environment through the joystick, such a task permitted individuals to detect the rewarded boxes by the relative position of the self, leading to a route knowledge”  Yes, I believe that this locomotion element is the best way to differentiate between the ‘route’ and ‘survey’ conditions,

L240, and further lines, “showed a not significant main effect” typo: should be written a ‘non-significant’?

L260-261 (L277-278) show the significant impact of familiarity upon ‘route’ and upon ‘survey’,

L268-269, “in addition, women made more errors than men only in the condition with low familiarity.” this is the key observation in the interaction of gender x familiarity, so requires post hoc analyses with correction in each of the three levels of familiarity, the reporting of effect sizes would also enable comparison with prior research,

276-277, whilst the global statistic reveals a significant gender difference, it would be informative to see if at each of the specific levels of familiarity there were significant gender differences, again this would underpin Discussion content and comparison with prior research,

L289-290, “The present paper was aimed at investigating gender differences in two spatial learning tasks” , an interesting way to label the tasks, I am not sure of the most appropriate way to label these tasks, but certainly, they are spatial learning tasks,

L297-299, “In particular, literature suggests that reaching more easily an high-order spatial knowledge lead men to be better in many spatial tasks; indeed, men are better at learning routes on a map both in the real world and in a virtual environment”,

the results could be interpreted differently to this argument,

the gender difference is present in the first phase of the route 'walking' in the survey 'non-walking' there needs to be further analyses to see if in any specific familiarity condition is associated with a gender difference. In some ways I would consider the non-walking condition as the format which directly identifies how well the participant accesses LTM and VSWM representation of the environmental array, whilst the walking condition further shows how the participant can use and operationalise this knowledge to direct the navigation,

L305-315, content well developed,

L328-329, there does appear to be a familiarity effect in the survey condition L293-295, and as highlighted above in my comments,

L330-331, “…and consequently they needed more attempts to reach a performance comparable to men. …”, this comment is not supported by the statistics as there is no interaction reported as with the route, and no between sex statistics at each level of familiarity are reported,

L345-366, content argument not supported in the data with an interaction effect driven by a larger sex difference at higher attentional demands (7 items to be located  in phase 3) in either task condition, route or survey

Author Response

Dear Reviewer 1,

please find attached the file including the point to point reply.

Thank you for the useful comments that we address along the revised manuscript

Best Regards

Laura Piccardi

Reviewer 2 Report

The manuscript entitled “The role of gender and familiarity in a modified version of the Almeria Boxes Room Spatial Task” by Bocchi and coworkers investigated the role of gender and familiarity in Route and Survey Tasks. The aim of the study was to fill the gap how gender and familiarity with the environment can affect spatial abilities when different modalities of environmental knowledge acquisition are required.

Their findings show that the tasks related to spatial learning along the route and types of navigation studies can be influenced by gender, with men showing better results. Importantly, the effect of familiarity modulated gender differences only in the route but not in the survey modality. In addition, knowing the environment can bridge the gender gap when the task is not cognitively demanding, as in the case of the Route task, while the difficulty of the task increases, as in the case of a Survey, familiarity is not sufficient to "fill in" the gender gap.

The authors speculate that the differences between the gender may be due to the involvement of different types of memory in men and women. In women, the prefrontal cortex is activated, which suggests the participation of working memory load in solving the navigation task, while men used the human navigation network. Thus, people with a more flexible mental representation of the environment are better at navigating tasks, even if they are very difficult.

On the whole, this study seems well conducted, methods are sound, data are convincing and the conclusions are in general supported by the data. However, there are a few issues, which need to be addressed:

  • In the Route Task the maximal trail duration was 150s. Whether in the Survey Task the maximal trail duration was the same?
  • Whether the selected students studied in the same field of study, or were students of different faculties? This information is missing from the manuscript.

Author Response

Dear Reviewer 2,

Thank you for the useful comments to improve the manuscript. 

Comments and Suggestions for Authors

The manuscript entitled “The role of gender and familiarity in a modified version of the Almeria Boxes Room Spatial Task” by Bocchi and coworkers investigated the role of gender and familiarity in Route and Survey Tasks. The aim of the study was to fill the gap how gender and familiarity with the environment can affect spatial abilities when different modalities of environmental knowledge acquisition are required.

Their findings show that the tasks related to spatial learning along the route and types of navigation studies can be influenced by gender, with men showing better results. Importantly, the effect of familiarity modulated gender differences only in the route but not in the survey modality. In addition, knowing the environment can bridge the gender gap when the task is not cognitively demanding, as in the case of the Route task, while the difficulty of the task increases, as in the case of a Survey, familiarity is not sufficient to "fill in" the gender gap.

The authors speculate that the differences between the gender may be due to the involvement of different types of memory in men and women. In women, the prefrontal cortex is activated, which suggests the participation of working memory load in solving the navigation task, while men used the human navigation network. Thus, people with a more flexible mental representation of the environment are better at navigating tasks, even if they are very difficult.

On the whole, this study seems well conducted, methods are sound, data are convincing and the conclusions are in general supported by the data. However, there are a few issues, which need to be addressed:

In the Route Task the maximal trail duration was 150s. Whether in the Survey Task the maximal trail duration was the same?   

     R. First of all the Authors wish to thank the Reviewer for providing suggestions for our work. We are glad that he/she found the paper convincing. For trial duration, the Reviewer is right. The maximal trial duration in the Survey task was 150s. We added this information in the description of the Survey task.

Whether the selected students studied in the same field of study, or were students of different faculties? This information is missing from the manuscript.   

     R. Students was enrolled from different field of study. We added this information in the Participants section.

Best Regards

Laura Piccardi